# Magnetic Nanoclusters Coated with Albumin, Casein, and Gelatin: Size Tuning, Relaxivity, Stability, Protein Corona, and Application in Nuclear Magnetic Resonance Immunoassay

**DOI:** 10.3390/nano9091345

**Published:** 2019-09-19

**Authors:** Pavel Khramtsov, Irina Barkina, Maria Kropaneva, Maria Bochkova, Valeria Timganova, Anton Nechaev, Il’ya Byzov, Svetlana Zamorina, Anatoly Yermakov, Mikhail Rayev

**Affiliations:** 1Laboratory of Ecological Immunology, Institute of Ecology and Genetics of Microorganisms of the Ural Branch of the Russian Academy of Sciences, Branch of PSRC UB RAS, 13 Golev str., 614081 Perm, Russia; kropanevamasha@gmail.com (M.K.); krasnykh-m@mail.ru (M.B.); timganovavp@gmail.com (V.T.); ivbyzov@gmail.com (I.B.); mantissa7@mail.ru (S.Z.); yermakov@imp.uran.ru (A.Y.); mraev@iegm.ru (M.R.); 2Department of Microbiology and Immunology, Biology Faculty, Perm State National Research University, 15 Bukirev str., 614000 Perm, Russia; i_barkina@mail.ru; 3Institute of Technical Chemistry of Ural Branch of the RAS, 3 Academician Korolev str., 614013 Perm, Russia; toxambj@gmail.com; 4M.N. Mikheev Institute of Metal Physics of the Ural Branch of the Russian Academy of Sciences, 18 S. Kovalevskoy str., 620108 Yekaterinburg, Russia; 5Ural Federal University named after the first President of Russia B.N. Yeltsin, 19 Mira str., 620002 Yekaterinburg, Russia

**Keywords:** nanoparticles, protein, assay, colloidal stability, antibody, protein G, streptavidin

## Abstract

The surface functionalization of magnetic nanoparticles improves their physicochemical properties and applicability in biomedicine. Natural polymers, including proteins, are prospective coatings capable of increasing the stability, biocompatibility, and transverse relaxivity (r2) of magnetic nanoparticles. In this work, we functionalized the nanoclusters of carbon-coated iron nanoparticles with four proteins: bovine serum albumin, casein, and gelatins A and B, and we conducted a comprehensive comparative study of their properties essential to applications in biosensing. First, we examined the influence of environmental parameters on the size of prepared nanoclusters and synthesized protein-coated nanoclusters with a tunable size. Second, we showed that protein coating does not significantly influence the r2 relaxivity of clustered nanoparticles; however, the uniform distribution of individual nanoparticles inside the protein coating facilitates increased relaxivity. Third, we demonstrated the applicability of the obtained nanoclusters in biosensing by the development of a nuclear-magnetic-resonance-based immunoassay for the quantification of antibodies against tetanus toxoid. Fourth, the protein coronas of nanoclusters were studied using SDS-PAGE and Bradford protein assay. Finally, we compared the colloidal stability at various pH values and ionic strengths and in relevant complex media (i.e., blood serum, plasma, milk, juice, beer, and red wine), as well as the heat stability, resistance to proteolytic digestion, and shelf-life of protein-coated nanoclusters.

## 1. Introduction

Magnetic nanoparticles find application in numerous fields of biomedicine, including drug delivery, tissue engineering, bioimaging, biosensing, and many others [1,2]. The magnetic nanoparticle-assisted isolation of cells, proteins, DNA, and other analytes from such complex matrices as whole blood, blood serum or plasma, urine, tissue extracts, food, and liquor are a powerful tool in biosensing, facilitating increases in analytical sensitivity and specificity [3]. Recently, magnetic nanoparticles have been extensively used in optical, electrochemical, and piezoelectric sensors, as well as in colorimetric point-of-care tests serving as tags, carriers of labels (e.g., enzymes), or both [4,5].

Nuclear magnetic resonance (NMR)-based biosensing is an area where magnetic nanoparticles are filling an extremely important role. The ability of magnetic nanoparticles to perturb the precession of nuclear spins in surrounding water protons stipulates their application in vivo as a magnetic resonance imaging (MRI) contrast [6] and in vitro as a nanoprobe for NMR-relaxometry [7]. There are two main approaches to in vitro NMR-assays. The first is a magnetic relaxation switch (MRSw), which relies on the aggregation of magnetic nanoparticles caused by the addition of an analyte. The aggregation/disaggregation of nanoparticles leads to a change in transverse relaxation time of protons (T2). The second approach is based on the binding of magnetic nanoparticles to the target and the removal of unbound particles. In this case, the relaxation time depends on the number of nanoparticles in the sample; this number is proportional to the analyte concentration. A detailed description of the principles underlying NMR sensors can be found in reviews [8,9,10]. NMR sensing platforms exploiting magnetic nanoparticles are used in food safety assessment; the diagnostics of sepsis, hemostatic disorders, and infectious diseases; and the detection of extracellular vesicles, tumor markers, enzymes, and ions [10,11,12]. Despite great progress in the field of magnetic sensing, improvements in the stability, relaxivity, and biocompatibility of magnetic nanolabels are of great importance for both in vivo and in vitro diagnostics [7,13,14].

Magnetic nanoparticles, a key element of NMR sensors, suffer from an instability in water solutions, the nonspecific sorption of molecules from the environment, and a lack of surface functional groups. Numerous surface coatings have been developed to address these challenges: polymers (e.g., dendrimers, polyvinylpyrrolidone, polyethylene glycol (PEG) and polyethyleneimine), zwitterions, small molecules (e.g., citrate), and inorganic materials (e.g., silica and gold) [6,15]. Proteins and polypeptides, as well as carbohydrates and lipids, are other prospective shielding agents to endow magnetic nanoparticles with stability in physiological media, biocompatibility, stealth properties, and multiple surface functional groups [16,17,18,19,20,21,22].

The goal of this study is to provide a direct comparison of the colloidal stability, functional activity, and physical-chemical properties of nanoclusters consisting of magnetic carbon-encapsulated iron nanoparticles coated with four proteins widely used in biomedicine: bovine serum albumin (BSA), casein, and gelatins A and B. We studied these nanoconjugates in the context of their application for in vitro assays, and more specifically, for T2 relaxometry. We chose the mentioned proteins for the following reasons:

(1) Serum albumin and casein coatings can increase the r2 relaxivity of magnetic nanoparticles due to their high hydrophilicity and ability to retain water molecules [13,23]. Gelatin has a strong affinity toward water molecules [24]; therefore, it is of great importance to reveal whether it can also increase relaxivity of magnetic nanoparticles.

(2) All proteins under study are extensively used for the preparation of nanoplatforms in drug delivery and theranostics because of their biocompatibility and excellent stability in physiological solutions [20,25,26]. The potential risks of human exposure to nanoparticles are still not well understood [27]; therefore, it is essential to ensure the biocompatibility of nanomaterials, even when they are designed for in vitro applications.

(3) BSA, gelatin, and casein are widely utilized as carriers for fluorophores, T1-contrast agents, enzymes, and therapeutics [20,25,26], therefore, the loading of a protein coating with agents providing multimodal biosensing is available.

(4) Albumin coatings decrease the nonspecific adsorption of serum proteins on the nanoparticles’ surface [17], facilitating a reduction in false-negative results for assays.

(5) Bovine serum albumin, casein, and gelatin are relatively inexpensive and widely available biopolymers.

To our knowledge, there remains a lack of comprehensive comparative studies of magnetic nanoparticles with different protein coatings. It is important that we examine the properties of protein-coated nanoclusters in parallel under the same conditions; we believe that only such an approach can provide an accurate comparison.

## 2. Materials and Methods 

### 2.1. Materials

Fe@C nanoparticles were synthesized as described elsewhere [28]. Sepharose CL-6B was from GE Healthcare (USA), nitrocellulose membrane SCWP, 8 μm pore size was from Merck (USA). Dialysis tubing with 10 kDa MWCO was from Thermo (USA). Bovine serum albumin (heat shock isolation), agarose was from Amresco (USA), casein (lot# C5890), gelatin type A gel strength 300 (lot# G2500), gelatin type B gel strength 75 (lot# G6650), biotinamidohexanoic acid N-hydroxysuccinimide ester, PEG 35 kDa, sodium dodecyl sulfate (SDS), β-mercaptoethanol, 4-aminobenzylamine, NaNO_2_ were from Sigma-Aldrich (USA). Streptavidin was from ProspecBio (Israel), recombinant protein G from *Streptococcus* sp. was kindly provided by Dr. Tatyana Gupalova, Institute of Experimental Medicine (St.-Petersburg, Russia). Tetanus toxoid was from Mikrogen (Russia), WHO standard of anti-tetanus IgG (TE-3) was from NIBSC (UK), sodium azide, ammonium persulphate, and glutaraldehyde (50%) were from AppliChem (Germany). Sodium hydroxide, glycerol, sodium chloride, sodium hydrogen phosphate, sodium dihydrogen phosphate, and Tween-20 were from Panreac (Spain). Bradford reagent, acrylamide, TEMED, Coomassie Brilliant Blue G-250, Bromphenol Blue were from Bio-Rad (USA), trypsin was from Samson-Med (Russia).

The following solutions were used: phosphate buffer solution (PBS, 0.15 M NaCl, 0.015 M Na_2_HPO_4_, 0.015 M NaH_2_PO_4_, and 0.1% NaN_3_, pH 7.25) and PBS-Tw (PBS + 0.1% Tween-20). A 25% (*v*/*v*) glutaraldehyde solution with pH 7.25 was prepared by mixing of 50% glutaraldehyde with PBS; pH and ionic strength were adjusted with 1 M NaOH and 5 M NaCl respectively. The solution of casein in water was prepared as follows: first, 2.5 mg of casein powder was diluted in 47.5 mL 1 M NaOH and dialyzed twice against 5 L of 0.1M NaCl and once against 5 L of deionized water; then casein was concentrated to 25–30 mL using PEG 35000 and centrifuged at 20,000× *g* for 100 min to remove the most of micelles (Appendix A). The final concentration of casein (8.8%) was determined using an extinction coefficient of 0.81 L·g^−1^·cm^−1^ [29]. All solutions were prepared using deionized water.

Blood serum samples were from volunteers aged from 21 to 58 years. All the procedures performed in the studies involving human participants were in accordance with the 1964 Declaration of Helsinki and its later amendments or comparable ethical standards. This research was approved by the Review Board of the Institute of Ecology and Genetics of Microorganisms UB RAS (IRB00010009). Written informed consent was obtained from the volunteers.

The following equipment was used: UV-VIS spectrophotometer Shimadzu UVmini 1240, an Asylum Research atomic-force microscope (United States), an MSE Soniprep 150 sonicator, a Malvern ZetaSizer Nano ZS particle analyzer, TGA/DSC1 (Mettler-Toledo) was employed to perform the thermogravimetric analysis (TGA), chromatography columns XK 16/40 and C 10/10, peristaltic pump P-1 were from GE Healthcare (USA), Mini Protean tetra cell and Mini-Sub GT cell for vertical and horizontal electrophoresis were from Bio-Rad.

A custom-made NMR relaxometer was used for the NMR-assay of antibodies and nanoparticle stability studies. The magnet assembly of the device was based on permanent SmCo magnets with a field of ~2 kOe (0.2 Tesla), with a heterogeneity of no more than 3 × 10^−5^ in the sample volume (1 cm^3^), that provides a measuring frequency of approximately 7.75 MHz. The reference frequency of the device was adjusted with an accuracy of 10 Hz to the Larmor resonance frequency of water protons in the sample to compensate for the temperature drift of the field before the measurement.

The device was based on the Analog Devices BF-937 DSP processor. Communication with a computer was carried out via USB; custom-made software was used to interact with a PC, through which the device was controlled. The same software was used for the mathematical processing of the results, including the calculation of the relaxation times.

The measuring unit was inserted into the magnet assembly of the NMR relaxometer and consisted of radio-frequency coils and capacitors in a 3D-printed plastic (PLA) case. Two different measuring units were constructed: one for liquid samples, from 10 to 100 µL, placed in the wells of a standard 96-well-striped ELISA plate, and the other one with a flattened coil (10 × 10 × 1 mm) for the NMR measurement of liquids in flat porous membranes. For the well-based measuring system, the SNR (signal-to-noise ratio) was 43 for a liquid volume of 10 μL and 378 for a liquid volume of 100 μL. For the planar system, the SNR was 15 for a liquid volume of 10 μL in the membrane; this value was more than enough to make reliable measurements of the relaxation times. 

### 2.2. Preparation of Aminated Fe@C (Fe@C-NH_2_)

First, 100 mg of Fe@C powder was added to 10 mL of 1 M HCl and left for 60 min at room temperature (RT), then washed five times with 10 mL of deionized water using magnetic separation and redispersed in 10 mL of 1 M HCl. Next, 120 µL of 4-aminobenzylamine (4-ABA) was added; afterwards, a glass tube with nanoparticles was placed at −20 °C for 30 min, then 70 mg of NaNO_2_ was added, and the resulting mixture was sonicated (6 mm probe, 100% amplification). When heating produced by the sonicator led to boiling and foaming, the sonication was interrupted to allow cooling. The total time of sonication was 60 min. Afterwards, the suspension was placed in 10 kDa MWCO dialysis tubing and dialyzed three times against 2 L of deionized water. The resulting suspension was transferred into the plastic tube and stored at +4 °C; then, 100 µL of 1 M HCl was added to prevent sedimentation of aminated nanoparticles. The concentration of Fe@C-NH_2_ was determined by thermogravimetric analysis (TGA) and can be found in the table (Appendix A).

### 2.3. Influence of pH, Ionic Strength and Protein-to-Nanoparticle Ratio on the Size of Protein-Coated Nanoclusters

Casein, BSA, and gelatins A and B were diluted in buffers with pH that ranged from 4 to 9 (pH 4 and 5—acetic buffer, pH 6, 7, and 8—phosphate buffer, and pH 9—borate buffer). The initial ionic strength of the buffers was 0.01 M, and it was adjusted to 0.15 and 0.5 M with 5 M NaCl where necessary. Fifty microliters of 10 mg/mL Fe@C-NH_2_ was added dropwise under vortex stirring, then water and 1 M NaOH/1 M HCl were added to make a final volume of 500 μL and a pH of 7.2–7.6. The resulting protein-to-nanoparticle mass ratios were 10:1, 5:1, 2.5:1, and 1.25:1. The suspensions were sonicated for 10 s (30% amplification, 3 mm probe), added while vortexing to an equal volume of 25% glutaraldehyde solution (see Section 2.1) and incubated for 30 min on a rotating mixer. For BSA and casein, all steps were carried out at room temperature, while the gelatin solutions and gelatin-coated nanoclusters were kept at +37–+40 °C to prevent gelation.

### 2.4. Synthesis of Protein-Coated Fe@C-NH_2_ Nanoclusters Conjugated with Streptavidin and Streptococcal Protein G

The general protocol for nanocluster conjugation with affine compounds is described in this section. The details of the synthesis procedure depended on the type of protein coating (BSA, casein, and gelatins type A and B), target nanocluster size, recognition molecule (streptavidin or protein G) and purpose (e.g., we did not add BSA, glycerol, or glycine to conjugates that were used in the UV-VIS, proteolysis, and protein corona studies) and can be found in Supporting Information. 

The suspension of Fe@C-NH_2_ (10 mg/mL) was added dropwise under vortex stirring into a protein solution. Where necessary, the pH was adjusted to 7.25 with 1 M NaOH, and the volume was adjusted with PBS. The final protein-to-nanoparticle ratio (mg:mg) was 5:1. The resulting suspension was added dropwise under vortex stirring to an equal volume of a 25% glutaraldehyde solution (see Section 2.1) and incubated on a rotating mixer (hereinafter, the rotation angle was set to 99°, and the rotation rate was 5 rpm) for 30 min. Then, the nanoclusters treated with glutaraldehyde were passed through a chromatography column to remove the excess of coating protein and glutaraldehyde. Fractions containing Fe@C-NH_2_/Protein nanoclusters were collected and concentrated in dialysis tubing (10 kDa MWCO, 2 mL/m) covered with a layer of 35 kDa PEG. In 2–3 h, the concentrated suspension was removed from the dialysis tubing and centrifuged at 1600× *g* for 5 min. The supernatant containing Fe@C-NH_2_/Protein nanoclusters activated with glutaraldehyde was added to a solution of the desired recognition molecule (streptavidin or protein G) in PBS under vortex stirring. The final recognition molecule-to-Fe@C-NH_2_/Protein ratios (μg:mg) were 10:1. 20:1, 40:1, 80:1, and 160:1. Conjugation was carried out overnight in a rotating mixer at +4 °C; glycine was added up to 6 mM to quench unreacted aldehyde groups, and the mixture was incubated at RT for one more hour. The excess of reactants was removed by gel-chromatography. The fractions with the highest concentration of nanoclusters were combined. Glycerol, BSA, and glycine were added to the final concentrations of 20%, 1%, and 6 mM, respectively. The conjugates were stored at +4 °C. The concentration of nanoclusters was determined by the absorbance at 450 nm, taking into account that the absorbance of a suspension of protein-coated nanoclusters with a known concentration (before the addition of glutaraldehyde) was preliminarily measured.

### 2.5. Assessment of Functional Activity of Protein-Coated Nanoclusters and Determination of Anti-Tetanus Toxoid in Serum Samples by NMR-Based Assay

The solid phase NMR-based immunoassay on nitrocellulose strips was used to assess the functional activity of different sizes of Fe@C-NH_2_/Protein/Str. Test-strips (5 mm × 60 mm) were cut from nitrocellulose with an 8 μm pore size. The test-strips were soaked in PBS for 5 min at +37 °C. Two microliters of biotinylated BSA (Bi-BSA) 10-fold diluted in PBS (from 100 μg/mL to 0.1 μg/mL) was spotted onto wet strips [30], which were further dried at +37 °C for 30 min and at RT for 15 min. Two microliters of 100 μg/mL BSA was spotted in the control zone. All subsequent steps were performed at +37 °C on an orbital shaker except for the measurement. The test-strips were blocked in 3.5 mL of 1% BSA in PBS-Tw, washed three times for 5 min in 5 mL of PBS-Tw, incubated in 3.5 mL of a 0.05 mg/mL solution of Fe@C-NH_2_/Protein/Str in PBS-Tw with 1% BSA for 60 min, and then washed 12 times in 5 mL of PBS-Tw. Parts of the test strip with spotted Bi-BSA were placed inside the coil of the relaxometer.

For the detection of the anti-tetanus toxoid IgG, a nitrocellulose membrane with a pore size of 8 μm was cut into 6 mm × 80 mm pieces and soaked in PBS for 5 min at +37 °C. Then, the wet membrane was incubated in 1.5 mL of a 20 μg/mL solution of tetanus toxoid in PBS for 30 min, dried successively at +37 °C for 30 min and at room temperature for 15 min, blocked in 1.5 mL of 1% BSA + 2% casein in PBS-Tw and triple washed with 1.5 mL of PBS-Tw for 5 min. Then, membranes coated with the tetanus toxoid were cut into 16 pieces (6 mm × 5 mm) (further referred to as “test strips”) which were used in subsequent procedures. The test strips were incubated in 400 μL of serum samples diluted 200-fold in PBS-Tw with 1% BSA and 2% casein for 60 min, washed three times with 600 μL of PBS-Tw and incubated in 400 μL of 0.05 mg/mL Fe@C-NH_2_/Protein/G (nanoclusters conjugated with protein G) in PBS-Tw with 2% casein for 60 min and washed 12 times. Solutions of the WHO anti-tetanus toxoid standard TE-3 in blocking buffer were used to make a calibration curve. All samples and calibrators were tested in triplicate. The measurement of T2 was performed by placing test strips inside the coil of the relaxometer. All assay steps were carried out at +37 °C on an orbital shaker except for the measurement.

### 2.6. Agarose Gel Electrophoresis

Agarose gel electrophoresis was used to study the size of nanoclusters and confirm the presence of a protein coating on their surface [31]. Samples (5 μL per well) were run in 0.5% agarose gel in 0.5 × TBE (Tris-borate) buffer, pH 8.3 at 75V. After that, gels were stained with staining solution containing 0.25 g Coomassie Brilliant Blue R250 in 100 mL of fixing solution: methanol:acetic acid:water = 4:1:4 and destained with fixing solution.

### 2.7. Protein Corona Study

In these experiments, Fe@C-NH_2_/Protein/Str conjugates were used without stabilizers (1% BSA and 20% glycerol), because they can interfere with the test results. The detailed preparation of the conjugates for these experiments is described in the Supporting Information. Different volumes of Fe@C-NH_2_/BSA/Str, Fe@C-NH_2_/Casein/Str, Fe@C-NH_2_/Gelatin B/Str, and Fe@C-NH_2_/Gelatin A/Str suspensions containing 100 μg of nanoclusters were centrifuged for 90 min at 10,000× *g*. The supernatant was removed, and 100 μL of whole human serum or plasma (pooled serum/plasma from several donors) was added to achieve a final concentration of nanoclusters of 1 mg/mL. To study protein adsorption on the noncoated aminated nanoparticles, 10 μL of 10 mg/mL Fe@C-NH_2_ was mixed with serum/plasma without prior centrifugation. The nanoparticles or nanoclusters were sonicated for 1 s (30% amplification, 3 mm probe), vortexed and kept on a rotator in a thermostat at +37 °C for 1 h. The fact that ultrasound can damage biopolymers is known [32], thus, we preliminarily confirmed that centrifugation and sonication did not affect the size and zeta potential of the nanoclusters. Unbound proteins were removed by washing with 1 mL PBS three times with centrifugation at 20,000× *g* for 15 min. The sedimented nanoclusters were mixed with 20 μL of sample buffer (0.5 M Tris-HCl pH 6.8, 5% β-mercaptoethanol, 10% SDS, 50% Glycerol, and 0.1% bromophenol blue), heated for 5 min at +95 °C and sonicated as described above. Ten microliters were loaded into the wells containing a 10% polyacrylamide gel, and then, the samples were run at 100 V per gel in Tris-HCl buffer until the proteins neared the end of the gel. The quantification of adsorbed proteins was carried out as follows. The samples were prepared in the same way except the proteins were eluted using 50 μL PBS containing 0.025% SDS and 4 M urea with heating (+95 °C, 5 min); the urea and SDS in mentioned concentrations do not interfere with the Bradford assay according to the manufacturer. The eluted serum proteins were separated by centrifugation (10,000× *g*, 15 min), and then, the protein concentration was determined by the Bradford assay (5 μL of undiluted sample + 250 μL Bradford reagent) using BSA to obtain a calibration curve. Samples without the addition of serum were used as a control. The concentrations of eluted proteins in the control samples were subtracted from those of the test samples. Three replicates were made for both test and control samples.

### 2.8. Thermal Stability of Nanoclusters

Forty microliters of Fe@C-NH_2_/Protein/G, supplemented with 1% BSA, 20% glycerol and 6 mM glycine, was incubated at temperatures ranged from +40 to +100 °C in a dry block thermostat for 60 min. The samples were diluted to 1:500 in PBS for the dynamic light scattering (DLS) measurements. Three replicates were made for each sample. The control samples were kept at room temperature throughout the experiment. 

### 2.9. UV-VIS Spectrophotometry

Suspensions of Fe@C-NH_2_/Protein/Str in PBS without stabilizers (BSA, glycine, and glycerol) were diluted in PBS, and their spectra were registered using PBS as a blank. Fe@C-NH_2_ was diluted in 0.1 M acetic buffer at pH 4 and measured against this buffer. 

### 2.10. Thermogravimetric Analysis

A total of 200 μL Fe@C-NH_2_ and Fe@C-NH_2_/BSA/Str samples were dried at 100 °C and placed in the TGA furnace. The measurement was carried out under air at a heating rate of 10 K·min^−1^ from 25 to 1000 °C.

### 2.11. Dynamic Light Scattering (DLS) 

The zeta potential and mean hydrodynamic diameter of the nanoclusters were measured by dynamic light scattering.

For size measurements, the samples were diluted to 1:500 (3 µL of nanoclusters suspension + 1500 µL of buffer) in buffers, which were preliminarily filtered through a polyethersulfone syringe filter with an average pore size of 0.2 µm to remove dust, and pipetted into plastic cuvettes (12 × 12 mm). Measurements were performed at a scattering angle of 173° in auto mode. A general purpose model was used to fit the data. The intensity-weighted size (main peak) is reported throughout the article unless otherwise stated.

For the zeta potential measurements, samples were diluted to 1:100 in deionized water (7 µL of nanoclusters suspension + 700 µL of water) in the same cuvettes. The measurements were performed in auto mode using a Dip Cell electrode (Malvern, UK). All measurements were done in triplicate.

### 2.12. Atomic Force Microscopy (AFM)

The samples of Fe@C-NH_2_/Protein/Str without stabilizers (BSA, glycerol, and glycine) were diluted with water to 10 μg/mL, and then, a drop of the sample suspension was placed onto a microscopic glass slide and dried at room temperature for 30–40 min.

### 2.13. Colloidal Stability of Nanoclusters

The assessment of colloidal stability utilized DLS, and the following buffers were used: 0.1 M acetate buffer (pH 4 and 5), 0.05 M sodium phosphate buffer (pH 6, 7, and 8), 0.1 M sodium borate buffer (pH 9 and 10). The ionic strength was adjusted to 0.15, 0.5, and 2 M using NaCl. All buffers were filtered through 0.2 μm polyethersulfone syringe filters and contained 0.1% sodium azide as a preservative. The samples of Fe@C-NH_2_/Protein/G were diluted to 20 μg/mL for the final volume of 750 μL in disposable plastic cuvettes. Three independent samples were prepared for each condition tested. After agitation for 60 min at room temperature, the size of the nanoclusters was measured by DLS. Then, the cuvettes with samples were left in a humidified chamber at room temperature, and repeated measurements were made in 24 h.

The colloidal stability was also assessed by NMR relaxometry. The samples of Fe@C-NH_2_/Protein/G were diluted to 10 μg/mL in 500 μL of the buffer, and then, 100 μL samples were added to the wells of a 96-well polystyrene plate and placed inside the relaxometer’s coil. The buffers and time points were the same.

### 2.14. Storage Stability of Protein-Coated Nanoclusters

The samples of Fe@C-NH_2_/Protein/G in PBS with 1% BSA, 20% glycerol and 6 mM glycine were stored at +4 and +37 °C (one sample per conjugate). Their hydrodynamic diameter was monitored throughout one month. The conjugates were diluted to 1:350 in 750 μL of filtered PBS. The size was measured in triplicate: three independent dilutions of each sample. 

Samples of Fe@C-NH_2_/Protein/Str conjugates from the size-tuning study were stored at +4 °C for 3–4 months, and their size was measured in the same way.

### 2.15. Stability of Nanoclusters in Complex Media

Red wine, orange juice, beer, and milk were purchased from the local supermarket. Pooled blood serum/plasma from three healthy individuals was thawed and centrifuged at 10,000× *g* for 15 min to remove aggregates. Fe@C-NH_2_/Protein/Str and Fe@C-NH_2_/Protein/G were diluted to 10 μg/mL in 500 μL of each medium, and then, 100 μL samples were added into the wells of a 96-well polystyrene plate and placed inside the relaxometer’s coil. Measurements were taken immediately upon the addition of nanoclusters and then after 1 and 5 h. The samples were kept at room temperature throughout the study. 

### 2.16. Stability of Nanoclusters to Proteolytic Digestion

The samples of Fe@C-NH_2_/Protein/Str without stabilizers (BSA, glycerol, and glycine) were diluted to 10 μg/mL with PBS (control samples) or with PBS containing 100 μg/mL of trypsin (test samples). The samples were kept at +37 °C in Eppendorf tubes (three tubes for each type of protein coating) in a dry block thermostat for 24 h. The size of the nanoclusters in test samples was measured from 5 to 30 min after the initial mixture and then after 1, 2, and 24 h; the control samples were measured after 2 and 24 h. 

### 2.17. Determination of Nanoparticles’ Relaxivity

The relaxation measurements were carried out using the homemade portable NMR relaxometer. The magnetic field was approximately 0.1 T; the operating frequency of the device was 4.317 MHz. The relative inhomogeneity of the magnetic field in the operating volume of the magnetic system (diameter of 10 mm and height of 10 mm) did not exceed 10^−4^. The spin-spin relaxation time (T2) was measured using the Carr–Purcell–Meiboom–Gill (CPMG) pulse sequence with echo spacing TE = 1 ms. Serial 10-fold dilutions of the magnetic nanoclusters were prepared. The starting concentration of the magnetic nanoclusters was approximately 0.02–0.03 mg/mL. After dilution, the nanocluster suspension was sonicated for 10 s (an immersion-type probe was used). Then, 100 μL of suspension was placed in a clean well of a 96-well culture plate (Thermo Fischer Scientific). The cell was placed in an axial measuring coil of the relaxometer. A linear relationship between relaxivity and the concentration of nanoclusters indicated the absence of agglomeration during the measurement.

## 3. Results and Discussion

### 3.1. Influence of pH, Ionic Strength, and Protein-to-Nanoparticle Ratio on the Size of Protein-Coated Nanoclusters

Conjugates of magnetic nanoparticles with proteins were synthesized using the following strategy. Fe@C Powder with a mean size of approximately 6 nm [33] was aminated. The aminated iron-carbon nanoparticles (Fe@C-NH_2_) were added to aqueous protein solutions. Two simultaneous processes occurred: a pH-dependent aggregation of Fe@C-NH_2_ and stabilization of aggregates by protein molecules (Fe@C-NH_2_/Protein). The formed surface protein layer was cross-linked with glutaraldehyde and then desired recognition molecules: streptavidin (Fe@C-NH_2_/Protein/Str) or protein G (Fe@C-NH_2_/Protein/G) was attached via the reaction of primary amines with carbonyl groups (Figure 1). It should be noted that cross-linking stabilizes nanoclusters, and their size depends only on the coating conditions.

In our previous study, we demonstrated that variations in pH could be applied to tune the size of BSA-coated nanoclusters [33]. Here, we investigated the influence of coating conditions (pH, ionic strength, protein-to-nanoparticle mass ratio and sonication time) on the interaction between different proteins (casein, BSA, and gelatins) and Fe@C-NH_2_ to obtain protein-coated nanoclusters of the desired size. Aminated nanoparticles were coated with proteins under a variety of conditions: pH ranging from 4 to 9, ionic strength ranged from 0.01 to 0.5, and protein-to-nanoparticle mass ratios from 10:1 to 1.25:1. The size of nanoclusters was measured in PBS after the glutaraldehyde cross-linking of the protein layer.

In our experiments, the size of nanoclusters resulted from a trade-off between the aggregation of Fe@C-NH_2_ at pH > 4 and the protein stabilization of Fe@C-NH_2_ aggregates. Fe@C-NH_2_ is stable at pH 4 [33] (see the zeta potential of Fe@C-NH_2_ in Appendix A) and aggregates slowly at pH 5 and 9, while at neutral pH, immediate aggregation occurs. Generally, a slight increase in size was observed when the coating was performed at pH 6–8; this effect was more pronounced for gelatin B (Appendix A). At high ionic strength, only the highest protein-to-nanoparticle ratios provided stability to the nanocomposites at pH between 6 and 8. The increase in ionic strength led to the overall growth of the nanoclusters’ size and polydispersity. Interestingly, in comparison with BSA and especially casein, the gelatin B coating efficiently stabilized nanoclusters, even at the highest salt concentration, without a sharp increase in the mean diameter. In total, a decrease in the protein-to-nanoparticle mass ratio caused the enlargement of the nanoclusters, especially when BSA was used. It is known that van der Waals forces, hydrogen bonding, Coulombic forces, and hydrophobic interactions are the main driving mechanisms of protein adsorption on nanoparticles [34]. Apparently, Coulombic forces possess a dominant role in the stabilization of Fe@C-NH_2_ by casein, because this protein provides good stability in solutions with lower salt concentrations. In contrast, the impact of the hydrophobic force prevails in the case of gelatin B. Gelatin A showed a poor ability to stabilize aminated nanoparticles even at the lowest ionic strength. Therefore, media with higher salt concentrations were not tested.

Sonication power is another parameter that can be used to manipulate the size of protein-coated nanoparticles [35]. Prolonged sonication allows up to a two-fold decrease in particles’ size and polydispersity (Appendix A).

In further experiments, we adjusted the synthesis conditions to obtain protein-coated magnetic nanoclusters with various sizes to explore their stability, relaxivity, and functional activity in the solid phase NMR-assay.

### 3.2. Synthesis and Relaxivity Study of Protein-Coated Nanoclusters with Tunable Size

The size of magnetic nanoparticles affects their r2 relaxivity [13] and may be critical for different aspects of immunoassays. Wang et al. demonstrated that the migration rate of magnetic nanoparticles inside the lateral flow test strip depends on their diameter, and in turn, has a dramatic impact on the assay duration and sensitivity [36]; furthermore, smaller magnetic nanoparticles provide a more efficient magnet-assisted pre-analytical enrichment of samples [3]. The application of larger nanoparticles and even microparticles in homogeneous NMR sensors can improve the lower limit of detection [37].

We prepared nanoclusters of iron-carbon magnetic nanoparticles coated with casein, BSA and gelatin B and conjugated with streptavidin. For each type of coating, three groups of conjugates were obtained: “small”, “medium”, and “large”. We should note that the size of each group depended on the coating type, e.g., we were unable to prepare Fe@C-NH_2_/Gelatin B/Str with a diameter less than 140 nm; at the same time, 100–110 nm nanoclusters can be easily obtained when the BSA coating is used. Despite the difference in absolute diameters, both conjugates were designated as “small”. Moreover, various streptavidin-to-nanoparticle ratios were used for all groups: from 10:1 to 160:1 (μg:mg), and in total, 45 conjugates were prepared. The main characteristics of conjugates are summarized in Table 1. Detailed information about the size, polydispersity, and zeta potential of each synthesized conjugate can be found in Appendix A. The size of Fe@C-NH_2_/Protein/Str was measured by DLS, while the migration of nanoclusters in agarose gel was used to characterize and compare the nanoclusters (Appendix A). The mobility of nanoclusters depends on their size and zeta potential; the nanocluster migration pattern confirms the results of DLS: casein- and BSA-coated nanoclusters with the highest zeta potentials moved strongly toward the positive electrode. “Large” nanoclusters hardly entered the gel, even 30 min after the start.

Interestingly, large nanoclusters can be seen in the wells in which “medium” nanoclusters were loaded (Appendix A). Moreover, “medium” nanoclusters distributed more diffusely along the path of migration compared with that of “small” nanoclusters, indicating the presence of a portion of large nanoclusters, and hence, a high polydispersity. However, according to DLS, the “medium” nanoclusters possess the lowest polydispersity. Moreover, it is well known that DLS overestimates the hydrodynamic diameter and shows a high polydispersity when large aggregates are present in the sample. A possible explanation is that nanoclusters larger than 170–180 nm do not enter the gel, while the smaller ones penetrate it. A diffuse band of nanoclusters represents those with a size between 130 and 170 nm, which are also present in the samples. Coomassie Blue staining of the protein-coated nanoclusters after gel-electrophoresis was performed to confirm the presence of a protein on the surface of Fe@C-NH_2_; unfortunately, the nanoclusters are too dark to detect any color change.

The functional activity of each Fe@C-NH_2_/protein/Str conjugate was examined in the solid phase NMR-assay. Biotinylated BSA was spotted on nitrocellulose test-strips in 10-fold dilutions, then, diluted conjugates were added, and the relaxation time depending on the number of nanoclusters attached to nitrocellulose was measured (Appendix A). Thus, we confirmed the applicability of the synthesized nanoclusters for NMR-assays; calibration curves were obtained for each conjugate (Appendix A). The type of coating, size of nanoclusters, and streptavidin-to-nanocluster ratio had a negligible effect on the shape of the calibration curves.

The protein coating of the magnetic nanoparticles can enhance their r2 relaxivity [13]. Recently, this effect was demonstrated for casein [38] and BSA [39,40]. Inspired by the above, we studied the r2 relaxivity of Fe@C-NH_2_/Protein/Str of different sizes. The relaxivity values of Fe@C-NH_2_/Casein/Str and Fe@C-NH_2_/Gelatin B/Str are independent of their size and are higher in comparison with that of Fe@C-NH_2_/BSA/Str (Figure 2). Moreover, the r2 relaxivity of Fe@C-NH_2_/BSA/Str was even lower than that of the parent Fe@C-NH_2_. We suggested that the different structures of the Fe@C-NH_2_/Protein/Str nanoclusters can be the reason for the differences in their r2 relaxivity values. The study of Fe@C-NH_2_/Protein/Str using transmission electron microscopy showed that individual Fe@C-NH_2_ particles are more uniformly distributed (Figure 3a) in the Fe@C-NH_2_/Casein/Str in comparison with Fe@C-NH_2_/Gelatin B/Str (Figure 3c) and especially Fe@C-NH_2_/BSA/Str (Figure 3b). Therefore, we suggest that the uniformity of the Fe@C-NH_2_ nanoparticle distribution inside the nanocluster, rather than the size of the Fe@C-NH_2_/Protein/Str nanocluster, affects the nanoclusters’ r2 relaxivity. Roca et al. [41] noted that the influence of magnetic nanoparticles on the proton relaxation rate is determined by not only the properties of an individual particle (saturation magnetization, nanoparticle size) but also how the magnetic nanoparticles are distributed inside their agglomerate. This effect is because the agglomerates of nanoparticles located inside the conjugate will be inaccessible to protons. In contrast, a uniform distribution of noninteracting magnetic nanoparticles inside a protein conjugate will make the greatest contribution to relaxation, provided that the medium separating them (in our case, protein) is permeable to protons.

Thus, Fe@C-NH_2_ nanoparticles are distributed nonuniformly in the protein matrix, affecting the Fe@C-NH_2_/Protein/Str relaxivity. The uniformity of distribution, in turn, depends on the nature of the protein and the procedure for the synthesis of Fe@C-NH_2_/Protein/Str. It should be noted that despite the Fe@C-NH_2_/Protein/Str from each size group (“small”, “medium”, and “large”) being prepared using the same portion of Fe@C-NH_2_/Protein, the resulting relaxivity varied significantly (in the range of 80 1/mM^−1^ × s^−1^) between conjugates from the same size groups (Table 1, Appendix A), reflecting the significant influence of fluctuations in the synthesis procedure on relaxivity. A further increase in the relaxation characteristics can be achieved by selecting such synthesis conditions that ensure the uniform distribution of individual nanoparticles in the protein shell and prevent the formation of their aggregates. It must be emphasized that the relaxivity of the synthesized nanoconjugates is comparable to the relaxivity of MRI contrasts based on iron oxides [23].

### 3.3. Characterization of Nanoclusters

The structure and morphology of nanoclusters were assessed using atomic force microscopy (AFM) and TEM. As shown in Figure 4a–d, Fe@C-NH_2_/Protein/Str has a nearly round shape. We did not assess the mean size of conjugates by microscopy because aggregates were produced during the sample preparation. The mean hydrodynamic diameters (intensity of the main peak) were determined by DLS, and they are reported throughout the article. The presence of proteins on the surface of Fe@C-NH_2_/Protein/Str and Fe@C-NH_2_/protein/G was confirmed by UV-VIS spectroscopy; there is a pronounced peak at 190–250 nm, indicating the successful functionalization (Figure 4f–i). 

Moreover, the changes in electrophoretic mobility (Appendix A) and dispersibility at different pH (Figure 4j) also prove the presence of the protein coating. Aminated iron-carbon nanoparticles immediately aggregate in the electrophoresis buffer, while the protein-coated nanoclusters enter the gel and move toward the positive electrode. Fe@C sedimented at pH 4 and 7, the Fe@C-NH_2_ are stable at pH 4 because of the electrostatic repulsion between protonated amino groups, but this is not the case at pH 7. Gelatin-coated nanoclusters are stable at both pH 4 and 7, and this is opposite to the behavior of casein- and BSA-coated ones, whose size remains unchanged at pH 7 but slowly grows at pH 4 (see Section 3.6).

Thermogravimetric analysis (TGA) was used to determine the concentration of Fe in Fe@C-NH_2_ and confirm the presence of a protein coating. The TGA curves of Fe@C-NH_2_, BSA, and Fe@C-NH_2_/BSA/Str (Figure 4e) showed that below 180 °C the weight loss of all samples was quite small (5–7%) because of the removal of physically and chemically adsorbed water. In the segment from 190 to 330 °C, the TGA curve of Fe@C-NH_2_ exhibits a gradual weight gain. This weight gain implies that the Fe cores are being slowly oxidized. The data also indicate that Fe@C nanoparticles are securely stable below 190 °C in air.

Above 330 °C, the full oxidation to Fe_2_O_3_ occurs, and the weight loss neglects the weight gain due to the oxidation of carbon to CO_2_. That is, the weight loss between 330 and 800 °C is recognized as an active chemical reaction in which the graphite phase is oxidized to CO_2_ gas. The oxidation temperature of the carbon shell in the Fe@C-NH_2_ nanoparticles is significantly lower than that of carbon nanotubes (527–727 °C) and bulk graphite crystals (846 °C); this outcome is ascribed to a mass of lattice defects resulting from the serious bending of the graphite atomic layers. From the TGA data, we calculate that the concentration of the magnetic Fe in a Fe@C-NH_2_ sample is 68.1 wt%. In contrast to Fe@C-NH_2_, there was no weight gain when the Fe@C-NH_2_/BSA/Str nanoclusters were analyzed, evidently due to the presence of the protein coating. The complete degradation of BSA took place in the range from 200 to 640 °C, and only Fe_2_O_3_ remained in the crucible after 650 °C. We calculated that the concentrations of magnetic Fe and BSA in the Fe@C-NH_2_/BSA/Str nanoclusters were 45.7 and 31.5 wt%, respectively.

### 3.4. NMR-Assay of Anti-Tetanus Antibodies

The protein-coated magnetic nanoclusters were applied to the solid phase NMR-assay of anti-tetanus antibodies. The concentration of antibodies correlates with the protection against tetanus and reflects the efficiency of the vaccination. According to the World Health Organization, the protective threshold of antitoxin concentration is 0.1 IU/mL [42]. Nitrocellulose test-strips were coated with tetanus toxoid and then treated with antibodies against tetanus toxoid. The antibodies were detected with protein G-conjugated magnetic nanoclusters, which decrease the relaxation of water protons inside the pores of the membrane. Thus, the T2 relaxation time of water protons is proportional to the antibody concentration (Figure 5).

Regardless of the material used to coat nanoclusters, they were applicable for antibody detection. Fe@C-NH_2_/BSA/G and Fe@C-NH_2_/Casein/G provided better analytical sensitivities; however, lower nonspecific signals and steeper slopes of dose-response curves were generated for gelatin-coated nanoclusters (Appendix A). We did not optimize the assay conditions but rather used the same assay parameters for all four tested conjugates, and there are clear differences in their performances. Generally, good day-to-day reproducibility of the analysis was obtained for all conjugates. However, significant variation was observed in the blank sample for Fe@C-NH_2_/Gelatin A/G (Appendix A). To assess whether the antibody in real samples can be detected, we tested 10 blood serum samples obtained from vaccinated individuals. Fe@C-NH_2_/Casein/G was used in the assay. The lower limit of detection was 0.52 mIU/mL (blank + 3 SD), which is less than that of double-antigen ELISA or bead-based multiplex immunoassay (Table 2). Nevertheless, analytical sensitivity of the assay was far less than the protective threshold and allows quantification of anti-TT IgG in unprotected individuals. A good correlation (r^2^ = 0.97) with the ELISA results was demonstrated. However, the NMR-assay significantly overestimated antibody concentrations, indicating that further optimization is necessary (Appendix A). 

### 3.5. Thermal Stability Study

The colloidal stability of magnetic nanoprobes at elevated temperatures is essential for applications in assays based on the isothermal amplification of DNA (e.g., loop-mediated isothermal amplification or rolling circle amplification) [49] or PCR [50], when reaction mixtures can be heated up to +95 °C. The aggregation of nanoparticles can negatively influence assay performance. Herein, we tested how heating affects the size of protein-coated nanoclusters. The interaction with the nanoparticles and cross-linking enhance the heat stability of proteins [51,52], and vice versa, where protein coatings can protect nanoparticles from heat-induced aggregation [53].

The size of Fe@C-NH_2_/BSA/G was almost the same until +80 °C; from +80 to +100 °C, the nanoparticle diameter sharply increased (Figure 6). Meanwhile, the diameter of Fe@C-NH_2_/Casein/G gradually increased from 132 to 144 nm, but the polydispersity indices of both conjugates did not exceed 0.2, reflecting good thermal stability. According to [54], irreversible heat denaturation of BSA starts at +50 °C and becomes more prominent with increasing temperature; in contrast, casein is an intrinsically unstructured protein with a better heat stability [55]. These differences can explain the nanoclusters’ behaviors upon heating. The hydrodynamic diameter of gelatin-B-coated nanoclusters did not change significantly. However, the polydispersity index increased at temperatures of +70 °C and higher. The increase in the polydispersity of Fe@C-NH_2_/Gelatin A/G occurred at +60 °C and higher. These data are in agreement with the findings of Bigi et al., demonstrating the denaturation of glutaraldehyde cross-linked gelatin gels at +70 °C [51]. In total, the thermal stability of protein-coated nanoclusters is appropriate for the application in homogeneous assays, which require heating. The nanoparticle size increase of 10–30 nm should not significantly influence the surface-to-volume ratio and relaxivity, especially for the casein and gelatin coatings (see Section 3.2). 

### 3.6. Colloidal Stability and Shelf Life

A resistance to aggregation is an inevitable requirement of nanoconjugates that is necessary for their practical implementation. Nanoparticles should be stable at neutral and near-neutral pH to be suitable for most of bioanalytical applications. Moreover, for some applications, a stability at extreme pH or ionic strength, at least short-term, is much needed. Herein, we directly compared the hydrodynamic diameter and polydispersity of magnetic nanoclusters coated with BSA, casein, and gelatins A and B after 1 and 24 h of incubation in buffers with pH 4–10 and a salt concentration from 0.15 to 2 M. Two different methods were used: dynamic light scattering and T2 relaxometry. The latter is sensitive to the size of the magnetic nanoclusters because their aggregation results in the growth of T2. This method is semiquantitative rather than quantitative and was only applied to strengthen the results obtained using DLS. 

Nanoparticles coated with BSA and casein were stable in solutions at pH 6–10 when the ionic strength did not exceed 0.5 M; however, aggregation was observed at pH 6 in 2 M PBS. A slight growth in the size of casein-coated nanoclusters can be seen after 24 h of incubation. This growth was more prominent in 2 M solutions, where nanocluster enlargement occurred even at alkaline pH. Importantly, there was no significant aggregation of gelatin-coated nanoclusters at any pH or ionic strength after 24 h. The increase in ionic strength from 0.15 to 2 M resulted in the overall growth of the nanoclusters’ diameter to approx. 20–30 nm, regardless of their coating. We should emphasize the fairly low reproducibility of the DLS measurements. Three technical replicates were prepared for each sample, and in some cases, a significant discrepancy between replicates was observed. Repeated measurements of the same cuvette sometimes gave opposing results (low/high polydispersity). Moreover, sometimes a dramatic decrease in size and polydispersity was observed for repeated measurements of the same replicate after 24 h of incubation, which is hardly possible. For these reasons, we excluded unreproducible results from the analysis used to prepare the graphs (Figure 7). Furthermore, the short-term stability of conjugates under the same conditions was assessed by T2 relaxometry (Figure 8). The results of the T2 measurements confirmed the previous findings. A significant increase in T2 was observed after the incubation of BSA- and casein-coated Fe@C-NH_2_ in buffers with acidic pH. The increase in T2 for casein-coated nanoclusters after 24 h was more prominent than that for Fe@C-NH_2_/BSA/G, especially in solutions with ionic strengths higher than 0.5 M. The T2 relaxometry was not sensitive enough to distinguish between the mentioned 20–30 nm size differences caused by the high salt concentration. However, a slight increase in T2 was found for gelatin-coated nanoclusters diluted in buffers with pH 4 and ionic strengths of 0.15 and 0.5 M, but not of 2 M. Some degree of aggregation of nanoclusters coated with gelatins A and B probably takes place at this pH, but we did not reveal it by DLS. 

Our observations are in contradiction with the findings of [56], who demonstrated the aggregation of glutaraldehyde cross-linked gelatin nanoparticles at pH 5. At the same time, Gaihre et al. [57] and Sivera et al. [58] respectfully obtained suspensions of Fe_3_O_4_ and Ag nanoparticles coated with gelatin that were stable at a pH range of 2–12. Gaihre et al. synthesized gelatin A- and gelatin B-coated iron oxide nanoparticles that were stable at pH 4–8. A slight sedimentation was observed for gelatin B-encapsulated nanoparticles but not for gelatin A-encapsulated nanoparticles at ionic strengths higher than 0.1. This result was explained by the more prominent “salting out” of the former provoked by more gelatin B adsorbed on iron oxide nanoparticles [59]. Furthermore, a gelatin A coating prevented gold nanoparticles from aggregation at high ionic strengths [60]. 

Several research groups also studied the stability of casein- and BSA-functionalized nanoparticles. Casein-encapsulated iron oxide nanoparticles were stable at pH 2–8 except for pH 4 [61]. Casein-coated zein nanoparticles were stable at I 0–200 mM and pH 2–9. However, a growth in diameter at pH 4 and 5 was observed [62]. A severe aggregation of casein-coated Ag nanoparticles was observed at pH lower than the isoelectric point of casein; however, this was not so at pH 8–13, and only a slight aggregation was observed at pH lower than 8. While, salt concentrations up to 0.5 M did not induce a change in their size [63]. BSA-coated quantum dot/magnetic nanoparticle composites were stable at ionic strengths up to 0.5 and at pH 5–13 but aggregated at pH 3 and 4 [64]. Zhang et al. demonstrated the good stability of quantum dots coated with BSA at ionic strengths up to 2 M and at pH ranging from 2–12 with small decreases at 5.6 and 6.1 [65]. BSA-coated magnetic nanoparticles were stable at pH 4–13 except for pH 5.6 and at an ionic strength from 10 mM to 1 M [66]. Gold nanoparticles with a BSA shell were also stable throughout pH 1.5–10 except at pH near 4.5–5 [67].

We can conclude that our results are consistent with those of previous studies; however, a direct comparison is complicated because of the different functionalization strategies. In our study, gelatin-coated magnetic nanoclusters were almost insensitive to pH variations in the range from 4 to 10, and BSA- and casein-coated nanoclusters aggregated at pH values near the isoelectric points of these proteins. Notably, such behaviors correspond with the zeta potential variations: the zeta potentials of Fe@C-NH_2_/BSA/G and Fe@C-NH_2_/Casein/G are between 0 and −20 mV at pH 4 and 5 (Figure 8). Surprisingly, there is no such distinct relationship between the zeta potential of gelatin-coated Fe@C-NH_2_ and its aggregation state. Moreover, despite the zeta potentials of both Fe@C-NH_2_/Gelatin A/G and Fe@C-NH_2_/Gelatin B/G being very low in comparison with the widely accepted ±30 mV stability threshold, they withstand aggregation, even at high ionic strengths, indicating the more significant role of hydration forces [68] and steric rather than the electrostatic repulsion between gelatin-coated nanoparticles [69]. Casein-coated nanoclusters are more sensitive to the increase in salt concentration. Therefore, we suppose that mainly the surface charge provides stability to these nanoclusters.

### 3.7. Storage Stability

The practical implementation of nanoparticle-based diagnostic reagents requires them to be stable for extended periods [70]. We monitored the size and polydispersity of Fe@C-NH_2_/Protein/G, which was stored at +4 and +37 °C for four weeks. A temperature range from +2 to +8 °C is widely used for the storage of bioreagents, including test kits and diagnostic reagents (e.g., enzyme conjugates). The exposure of components of test kits to elevated temperatures is a rapid and efficient way to assess their long-term stability. According to [71], seven days at +37 °C is equivalent to one year at +4 °C. Apparently, this approximation applies mainly to proteinaceous reagents, which are liable to thermal denaturation. However, nanoparticles aimed for use in in vivo applications are also tested under accelerated conditions [72].

The hydrodynamic diameter and polydispersity of protein-coated nanoclusters did not alter after one month of the storage. The most notable change was observed for Fe@C-NH_2_/Casein/G stored at +4 °C, in which the size was increased from 128 to 142 nm (Figure 9). At the same time, elevated temperatures did not affect the size of this conjugate. Therefore, the growth of nanoclusters at +4 °C was rather due to the variability in the measurements. To reinforce these results, we assessed the diameter and polydispersity alterations of the nanoclusters obtained during the course of the size-tuning experiments, which were stored for four months (Appendix A). Generally, there were no systematic changes in the hydrodynamic diameters. A time-dependent size alteration in some samples (denoted with arrows in Appendix A) can be attributed mainly to the variability of the DLS method because not only increases but also decreases in the mean diameter after storage were observed.

### 3.8. Stability of Nanoclusters in Complex Media

Magnetic nanoparticles are widely used in the homogeneous assays of pathogens, toxins, heavy metals, and other analytes in food, physiological, and environmental samples. In the course of these assays, the nanoparticles encounter a complex mixture of proteins, carbohydrates, lipids, and their complexes, which can induce nanoparticle aggregation and the subsequent loss of functions. The nonspecific aggregation of nanoparticles is undesirable for aggregation-based assays (e.g., T2 relaxometry) because it leads to false-positive results. Moreover, the aggregation of nanoparticles can reduce the efficiency of the magnetic enrichment [73]. We studied the stability of protein-coated nanoclusters in blood serum/plasma, orange juice, milk, red wine, and beer by T2 relaxometry. Plasma and serum are among the most significant specimens in clinical diagnostics; milk, juice, beer, and wine are popular foods that are usually tested for the presence of pathogens and their toxins before becoming available to consumers. The tested media do not contain any magnetic material, and hence, do not affect the results of the T2 relaxometry, which depends only on the aggregation of magnetic nanoparticles diluted in these media [74]. Most of the nanoparticle-based assays require only minutes- to hours-long incubations of nanoprobes in the samples under study. Therefore, we assessed the T2 change of diluted nanoclusters after 1 and 5 h of incubation.

There was no significant change in T2 after the incubation of casein- and BSA-coated nanoclusters in juice, milk, and blood serum/plasma. An increase in T2 was observed for all four types of nanoclusters diluted in beer. However, for Fe@C-NH_2_/BSA/G and Fe@C-NH_2_/Casein/G, a sharp increase in T2 was observed for the gelatin-coated nanoclusters after dilution in serum and plasma. Moreover, aggregates of gelatin-encapsulated nanoclusters in blood samples were visible by the naked eye. The aggregation of gelatin-coated nanoclusters was not associated with the interaction between protein G and the IgG from the plasma/serum because the experiment with Fe@C-NH_2_/Protein/Str produced the same result (Figure 9). All tested conjugates were aggregated in red wine. The complex relationship between the size and surface structure of nanoparticles and their relaxivity is well known [13]. Thus, we cannot fully explain the alteration of T2 in each case. Nevertheless, the aggregation of nanoclusters and adsorption of the media’s components on their surface definitely have a significant impact. According to the obtained results, we can conclude that gelatin-coated nanoclusters are inappropriate for the determination of analytes both in undiluted serum and plasma. All four conjugates can be added to whole milk and juice. 

### 3.9. Protein Corona

The blood serum composition represents a variety of pathological and physiological processes that makes serum one of the most informative and important clinical specimens. Nonspecific interactions of nanoparticles with serum components are inevitable when their mixing takes place during the assay procedure. Matrix interference is a significant issue in magnetic nanoparticle-based biosensing, causing a dramatic decrease in specificity [75]. According to recent reports, protein shells can provide antifouling properties by decreasing the amount of nonspecific binding of cells and medium components, first of all, serum proteins [65,76,77]. At the same time, the coating of nanoparticles with biocompatible natural polymers can increase the nonspecific protein binding [78]. Herein, we assessed the adsorption of serum proteins on the surface of Fe@C-NH_2_/Protein/Str by SDS-PAGE and Bradford assays.

Different protein coronas are formed after the incubation of nanoclusters in serum and plasma (Figure 10 and Appendix A). The overall amount of adsorbed protein from blood serum was the same for both coated nanoclusters and parent uncoated nanoparticles (Fe@C-NH_2_) except for Fe@C-NH_2_/Gelatin A/Str, which was bound with twice as much protein. However, we should note that direct comparisons of protein binding capacity are not completely correct because protein quantification by Coomassie staining depends on the amino acid composition of the protein under study. We find explaining the increased serum protein binding by gelatin-A-coated nanoclusters to be quite challenging, especially in the light of the fact that gelatins A and B have the same protein binding pattern and that the protein bands in stained gels have the same intensity. Moreover, according to SDS-PAGE, a significantly higher adsorption of plasma proteins by Fe@C-NH_2_/Gelatin B/Str was revealed (Figure 10). The protein coating did not reduce the nonspecific binding of serum proteins but rather led to a change in the composition of the protein corona. This fact corresponds with the results of Mirshafiee et al., who demonstrated that proteins could promote the adsorption of certain groups of serum proteins and reduce the adsorption of others in comparison with uncoated nanoparticles [79]. Additionally, smaller nanoparticles bind less protein from serum/plasma both per nanoparticle [80] and per unit area [81]. Uncoated Fe@C-NH_2_ nanoparticles have the lowest hydrodynamic diameter; however, they rapidly aggregate upon mixing with serum, and an accurate determination of their diameter is complicated. Gelatin A-coated nanoclusters are the largest one, but their size is only 30 nm larger than that of Fe@C-NH_2_/Gelatin B/Str (186 nm vs. 154 nm). At the same time, Fe@C-NH_2_/BSA/Str (111 nm) and Fe@C-NH_2_/Casein/Str (119 nm) are approximately 30–40 nm smaller than Fe@C-NH_2_/Gelatin B/Str, but they all adsorb the same amount of protein.

### 3.10. The Resistance of Nanoclusters to Proteolysis

Proteolytic cleavage by proteases from the environment reduces the shelf-life of proteins and their conjugates [82]. To assess the impact of protease digestion on the structural integrity of protein-coated magnetic nanoclusters, the latter were mixed with a trypsin solution, and the Fe@C-NH_2_/Protein/Str size was monitored within 24 h. We used trypsin at high concentration (100 μg/mL), similar to that of the intestinal secretion of a human adult (approx. 100–500 μg/mL) [83], to provide an accelerated study.

An almost two-fold drop in the mean diameter of gelatin-coated nanoclusters was observed after 1 h of incubation (Figure 10) without further changes. Conversely, only a slight decrease in BSA- and casein-coated nanocluster diameters occurred. After 24 h of agitation at +37 °C, the size of the nanoclusters increased in the control samples (without trypsin) but not in the test samples. In the previous sections, we noted that all conjugates were stable to heating; however, in this experiment, highly diluted (10 μg/mL) suspensions were used without the addition of stabilizers including BSA, glycerol, and glycine. These results demonstrate that stabilizers are essential for the preservation of conjugates’ properties. Interestingly, after the trypsin treatment, the Z-average diameter of the gelatin-coated nanoclusters was approximately 60–80 nm, which is equal to that of the parent Fe@C-NH_2_. A possible explanation for the different stabilities of coatings to proteolytic digestion is given below.

According to [84], the trypsin degradation rate of albumin nanoparticles covalently linked with glutaraldehyde is inversely proportional to the cross-linking degree, and only 9% of nanoparticles degraded within 24 h. Similar results were obtained for the ionically cross-linked casein nanoparticles [85]. In our work, we used a great molar excess of glutaraldehyde capable of involving all available primary amine groups in cross-linking reactions. Gelatins A and B contain the lowest amount (3.29% and 4.01%) of lysine [86] in comparison with that of casein (8.10%) and BSA (15.30%) [87,88]; therefore, fewer cross-links are formed between the primary amines of gelatin molecules [89]. Other factors such as conformation differences among the adsorbed proteins and their availability to proteolytic cleavage can also contribute to higher/lower resistance of nanoparticles to trypsinolysis. For example, Cao et al. demonstrated that adsorption on titanium dioxide nanoparticles can protect casein against gastric and intestinal digestion [90], and gold nanoparticles coated with proteins were stable for several days in PBS containing a proteolytic enzyme [91].

## 4. Conclusions

Protein coatings endow nanoparticles with several favorable properties, enhancing their performance in various biomedical applications. In the present study, we synthesized nanoclusters of magnetic iron-carbon nanoparticles coated with four proteins (BSA, casein from bovine milk, and gelatins A and B), examined their physical-chemical properties from the perspective of applications for in vitro diagnostics and used nanoclusters conjugated with Streptococcal protein G as labels in an NMR immunoassay of IgG against the tetanus toxoid. The developed immunoassay allows for the determination of protection against tetanus in patients vaccinated with the diphtheria-tetanus-pertussis vaccine.

All types of protein coatings provide the nanoclusters with an excellent long-term storage stability. The gelatin layer prevents nanoclusters from aggregating over a wide range of pH: from 4 to 10, whereas casein- and BSA-coated nanoclusters are stable at pH 6–10. The protein-coated nanoclusters withstand salt concentrations up to 2 M without a significant change in size. We demonstrated that protein-coated magnetic nanoparticles maintain their size in complex media (juice, milk, serum, and plasma) and at elevated temperatures, facilitating their applications in homogeneous assays of biomarkers and foodborne pathogens, as well as their analyses based on the amplification of nucleic acids. The type of protein coating and size of nanocluster do not significantly affect the r2 relaxivity of nanoclusters or their performance in NMR-assays. Generally, the transverse relaxivity of protein-coated nanoclusters was between 200 and 350 1/mM^−1^ × s^−1^. 

We suppose that the developed protein-coated nanoclusters can be applied for not only in vitro but also in vivo diagnostics (e.g., as a T2 contrast in MRI) due to their good stability in physiological media and high relaxivity (in comparison with clinically approved nanolabels) [23]. The proteins are biocompatible and biodegradable; moreover, the protein layer can be loaded with therapeutics or various labels, providing multimodal imaging.

## Figures and Tables

**Figure 1 nanomaterials-09-01345-f001:**
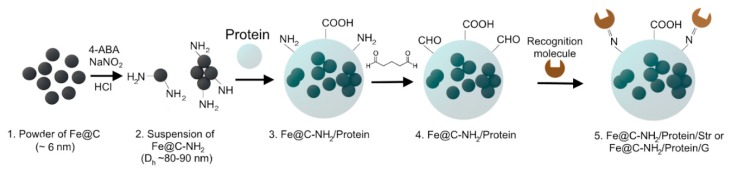
Synthesis of Fe@C-NH_2_/Protein/Str and Fe@C-NH_2_/Protein/G. 4-ABA-4-aminobenzylamine.

**Figure 2 nanomaterials-09-01345-f002:**
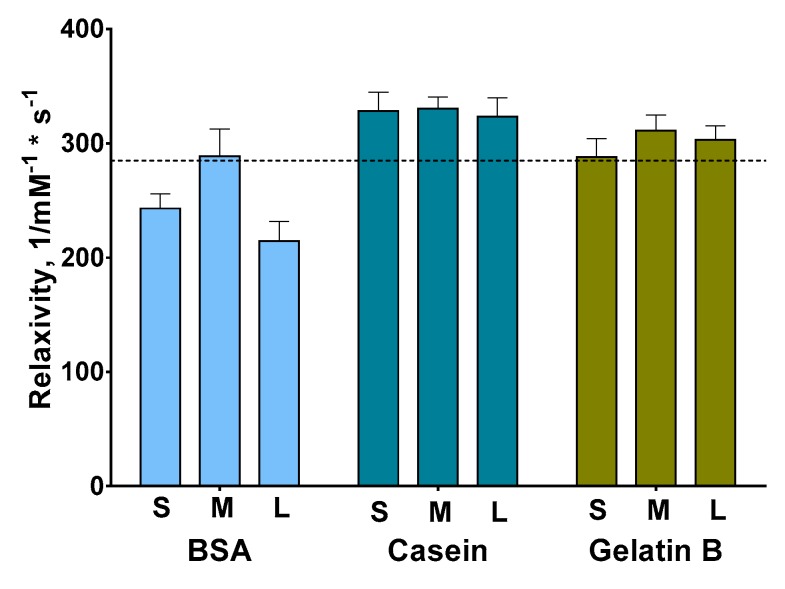
Relaxivity of conjugates with different sizes and coating types, *n* = 5, mean ± SD. Letters “S”, “M”, and “L” indicate “small”, “medium”, and “large” nanoclusters, respectively; the dashed line indicates the relaxivity of the parent Fe@C-NH_2_ (285 1/mM^−1^ × s^−1^). Conjugate Fe@C-NH_2_/Gelatin B/Str with highest relaxivity was excluded when mean relaxivity values were calculated.

**Figure 3 nanomaterials-09-01345-f003:**
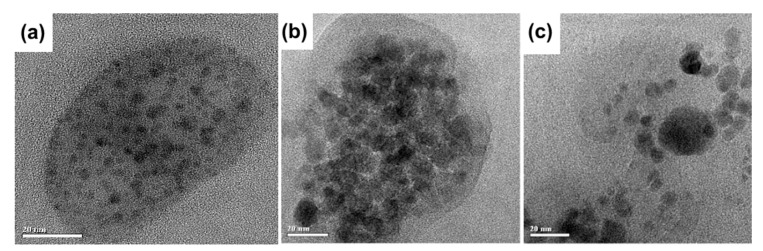
TEM images of Fe@C-NH_2_/Casein/Str (**a**), Fe@C-NH_2_/BSA/Str (**b**), and Fe@C-NH_2_/Gelatin B/Str (**c**). Scale bars are 20 nm.

**Figure 4 nanomaterials-09-01345-f004:**
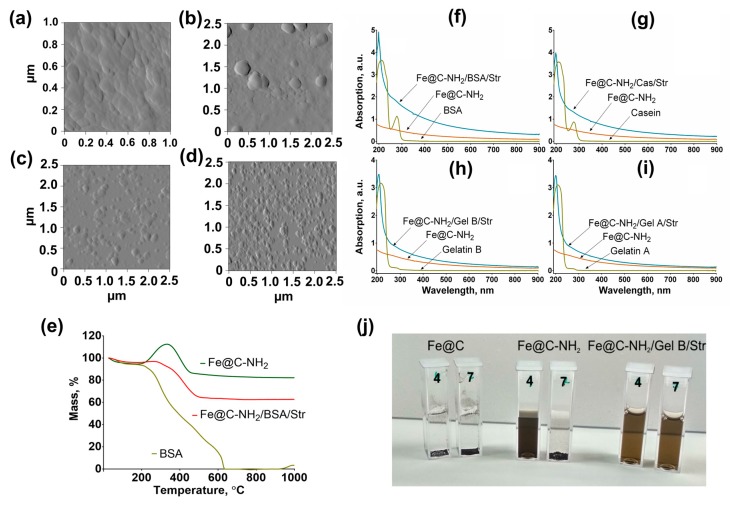
Properties of nanoclusters coated with different proteins. Upper left: atomic force microscopy (AFM) images of nanoclusters coated with different proteins: (**a**) BSA, (**b**) casein, (**c**) gelatin A, and (**d**) gelatin B; lower left: (**e**) thermogravimetric analysis (TGA) curves of Fe@C-NH_2_, BSA and Fe@C-NH_2_/BSA/Str in airflow; upper right: UV-Vis spectra of Fe@C-NH_2_, Fe@C-NH_2_/Protein/Str and proteins: (**f**) BSA, (**g**) casein, (**h**) gelatin B, and (**i**) gelatin A; and lower right: (**j**) stability of Fe@C, Fe@C-NH_2_, Fe@C-NH_2_/Gelatin B/Str in buffers with pH 4 and 7.

**Figure 5 nanomaterials-09-01345-f005:**
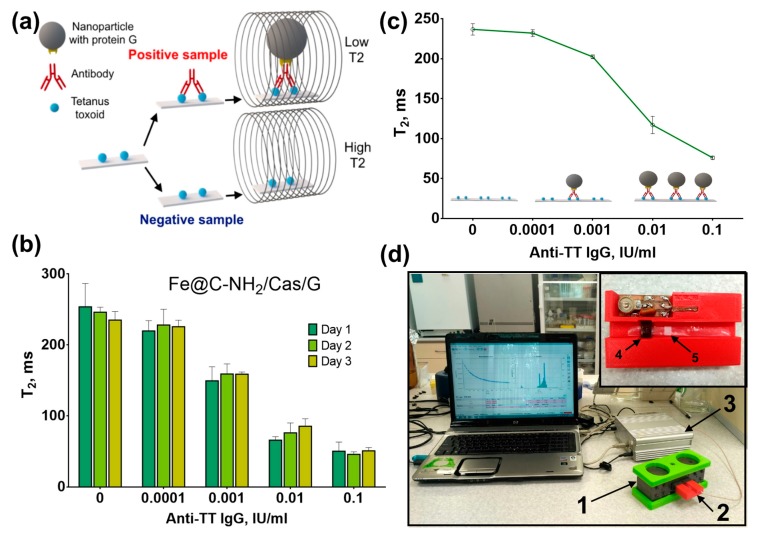
Application of protein-coated magnetic nanoclusters in nuclear magnetic resonance (NMR)-immunoassay of IgG against the anti-tetanus toxoid. (**a**) the principle of the assay; (**b**) day-to-day variability of the anti-TT NMR-assay; (**c**) dose-response curve obtained using Fe@C-NH_2_/Casein/G; (**d**) NMR-relaxometer and sample holder (inset): 1—magnet, 2—sample holder, 3—NMR-relaxometer, 4—radio-frequency coil, and 5—test-strip in plastic envelope.

**Figure 6 nanomaterials-09-01345-f006:**
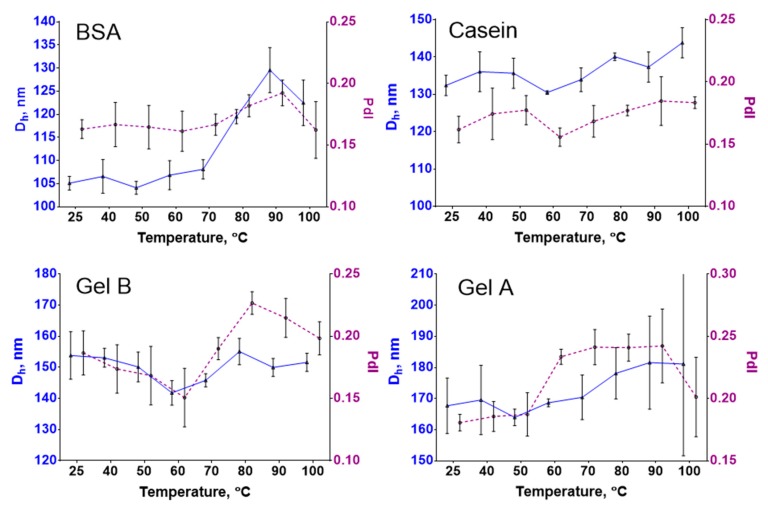
Thermal stability of (**BSA**) Fe@C-NH_2_/BSA/G; (**Casein**) Fe@C-NH_2_/Casein/G; (**Gel B**) Fe@C-NH_2_/Gelatin B/G; (**Gel A**) Fe@C-NH_2_/Gelatin A/G. Solid line—hydrodynamic diameter; dashed line—polydispersity index, *n* = 3, mean ± SD.

**Figure 7 nanomaterials-09-01345-f007:**
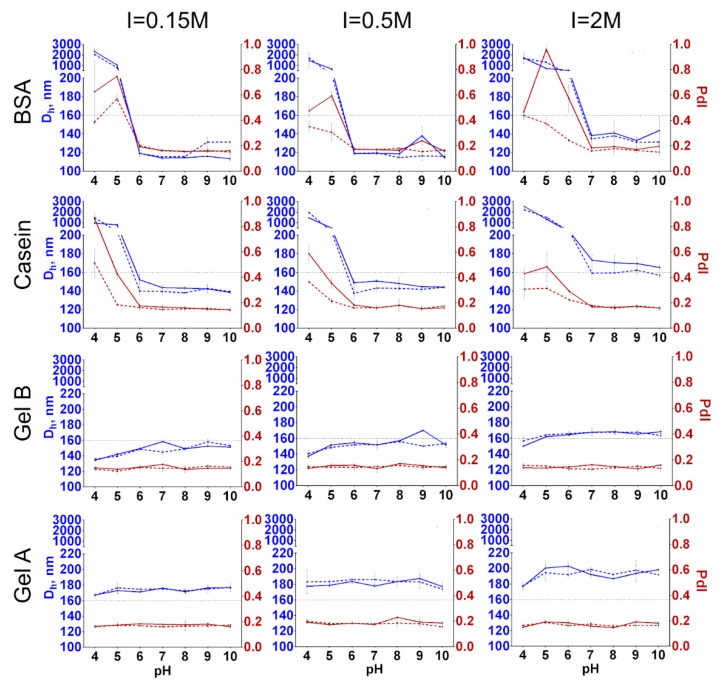
Colloidal stability of Fe@C-NH_2_/Protein/G in buffers with different pH and ionic strength values. Ionic strength values are specified at the top of the figure. “BSA”, “Casein”, “Gel B”, and “Gel A” indicate coating protein. Blue line—hydrodynamic diameter at 0 (dashed line) and 24 h (solid line); orange line—polydispersity index at 0 (dashed line) and 24 h (solid line), *n* = 3, mean ± SD.

**Figure 8 nanomaterials-09-01345-f008:**
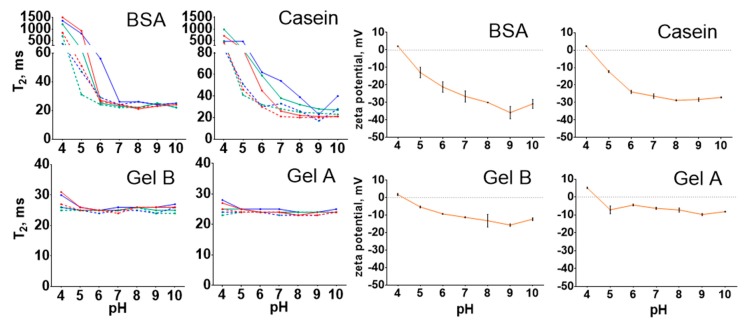
The T2 relaxation times of Fe@C-NH_2_/Protein/G diluted in buffers with different pH and ionic strength values and the zeta potentials of Fe@C-NH_2_/Protein/G. “BSA”, “Casein”, “Gel B”, and “Gel A” indicate coating protein. T2 in 0.15 M (red line), 0.5 M (green line), and 2 M (blue line) buffers at 0 (dashed line) and 24 h (solid line), *n* = 3, mean ± SD.

**Figure 9 nanomaterials-09-01345-f009:**
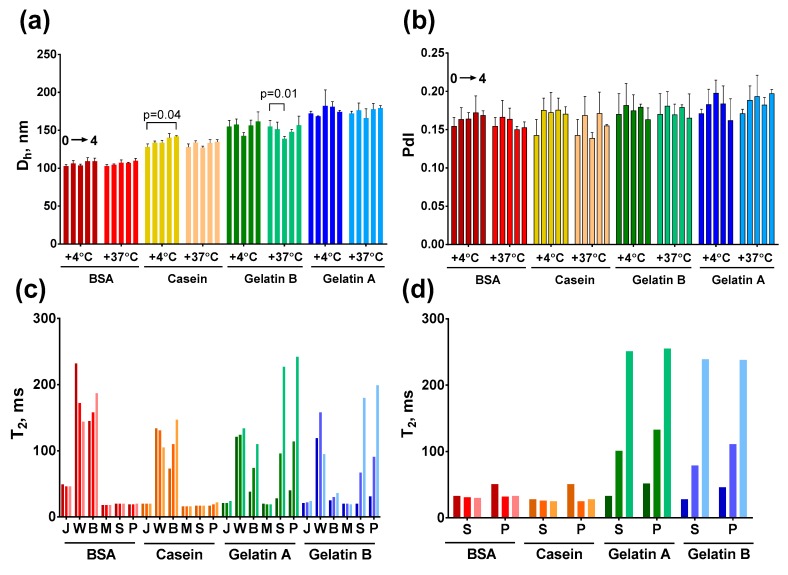
Storage stability of Fe@C-NH_2_/Protein/G and stability of Fe@C-NH_2_/Protein/G and Fe@C-NH_2_/Protein/Str in complex media. Upper row: changes in hydrodynamic diameter (**a**) and polydispersity index (**b**) during four weeks of storage at different temperatures; the five bars represent the size or PdI at week 0, 1, 2, 3, and 4 (from left to right), statistics: two-way ANOVA with Dunnet’s post-hoc test, *n* = 3, mean ± SD; lower row: the T2 of Fe@C-NH_2_/Protein/G (**c**) and Fe@C-NH_2_/Protein/Str (**d**) diluted in juice (J), wine (W), beer (B), milk (M), blood serum (S), and plasma (P); the three bars represent T2 at 0, 1, and 5 h (from left to right).

**Figure 10 nanomaterials-09-01345-f010:**
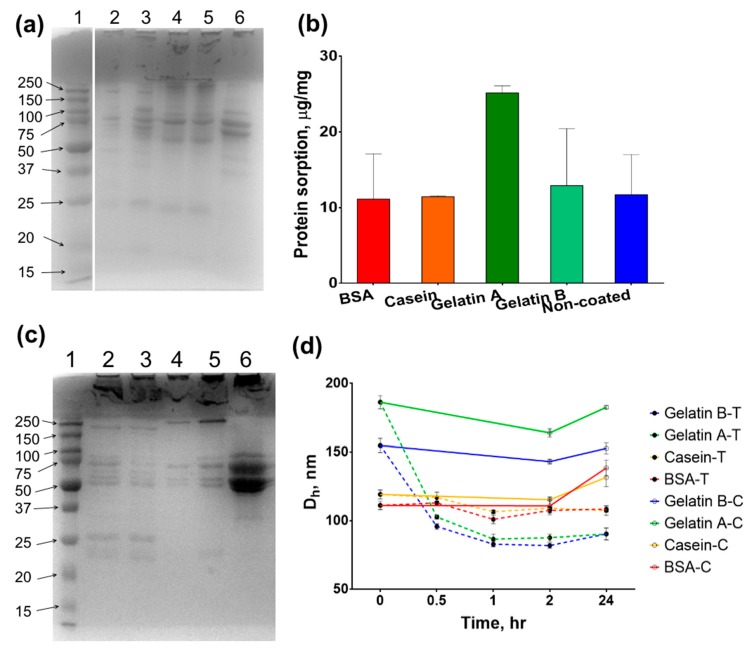
Protein corona and stability of nanoclusters to proteolysis. Protein coronas of Fe@C-NH_2_/BSA/Str (lane 2), Fe@C-NH_2_/Casein/Str (lane 3), Fe@C-NH_2_/Gelatin A/Str (lane 4), Fe@C-NH_2_/Gelatin B/Str (lane 5), and Fe@C-NH_2_ (lane 6) in blood serum (**a**) and plasma (**c**), with lane 1-protein markers (kDa). (**b**) The sorption of serum proteins on Fe@C-NH_2_/Protein/Str and Fe@C-NH_2_. (**d**) The size of Fe@C-NH_2_/Protein/Str after incubation in the trypsin solution (dashed line, filled circles) or PBS (solid line, open circles). Coating: BSA (red), casein (yellow), gelatin A (green) and B (blue), *n* = 3, mean ± SD.

**Table 1 nanomaterials-09-01345-t001:** Properties of conjugates prepared during size-tuning experiments.

Coating	Fe@C-NH_2_/BSA/Str			Fe@C-NH_2_/Casein/Str			Fe@C-NH_2_/Gelatin B/Str		
Group	“Small”	“Medium”	“Large”	“Small”	“Medium”	“Large”	“Small”	“Medium”	“Large”
**D_h, nm_**	114–121	172–186	211–233	114–131	194–198	235–274	142–160	212–243	279–309
**PdI**	0.188–0.208	0.158–0.175	0.247–0.258	0.178–0.201	0.132–0.169	0.202–0.243	0.211–0.248	0.200–0.260	0.227–0.257
**Zeta potential, mv**	−24–−25	−23–−25	−24–−25	−25–−27	−25–−26	−24–−26	−10–−12	−9–−10	−9–−10
**Relaxivity, 1/mM^−1^ × s^−1^**	234–265	262–322	186–230	310–354	318–340	303–345	266–306	296–377	282–324

Note: Severe aggregation was observed in the conjugate Fe@C-NH_2_/Gelatin B/Str synthesized at a streptavidin-to-nanocluster ratio of 10:1 immediately after the synthesis, and this conjugate was excluded from our study.

**Table 2 nanomaterials-09-01345-t002:** Comparison of the NMR-assay with anti-TT IgG detection methods described in literature.

Assay	Label	LOD, mIU/mL	Assay duration, h	Reference
ELISA	horseradish peroxidase	0.01	4	[43]
Multiplex immunoassay	fluorescent beads	0.01	1.25	[44]
Lateral flow assay	gold nanoparticles	10	0.25	[45]
Surface plasmon resonance assay	gold nanoparticles	5	2.15	[46]
ELISA on polymer fibers	horseradish peroxidase	0.5	2.15	[47]
Microfluidic assay	fluorescent dye	100	Less than 1	[48]
NMR assay	carbon-coated iron nanoparticles	0.52	3–4	This work

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
