# Peer review of "Magnetic Nanoclusters Coated with Albumin, Casein, and Gelatin: Size Tuning, Relaxivity, Stability, Protein Corona, and Application in Nuclear Magnetic Resonance Immunoassay"

_nanomaterials, 2019, doi:10.3390/nano9091345_

Round 1

Reviewer 1 Report

This paper seems to be well organized and written with plenty of data. These method can be applied for not only in vitro but also in vivo due to their excellent physical and chemical characteristics using protein-coated nanoclusters. However, some minor issues and comments should be clarified before publication.

1. In Fig. 6., are there any non-specific binding (NSB) of protein-coated magnetic nanoclusters in NMR-immunoassay of anti-tetanus antibodies with negative samples. The reviewer thinks that we can see NSB phenomena in high sensitive assay format. In addition, Could you give an explanation about the results of Fig.6(b) of slight increase of T2 with day to day variability?

Author Response

Dear Editor-in-Chief and Referees

Authors are very grateful for your comments to our manuscript and thoughtful suggestions. Based on these comments and suggestions, we have made modifications to the original paper. We believe that the manuscript has been greatly improved and hope it has reached your journal's standard. Please, find our answers to yours comments below. Added or changed pieces of text are highlighted with yellow color.

We changed the title of the article because the NMR-assay is not sensor, but rather a dot-immunoassay with disposable test-strips and NMR detection.

Reviewer #1

Comment

In Fig. 6., are there any non-specific binding (NSB) of protein-coated magnetic nanoclusters in NMR-immunoassay of anti-tetanus antibodies with negative samples. The reviewer thinks that we can see NSB phenomena in high sensitive assay format.

Response

We did not observed any non-specific binding (NSB) in negative samples. We tested negative rabbit serum (negative human serum are difficult to obtain because of high DTP vaccination coverage in our region) diluted 1/50 in blocking buffer and blank sample (blocking buffer) and measured T2 was the same. NSB was observed when casein was not added in blocking buffer (we described the same effect in our previous work: DOI 10.1021/acs.langmuir.8b01255, page 10325).

Comment

In addition, Could you give an explanation about the results of Fig.6(b) of slight increase of T2 with day to day variability?

Response

We performed two-way ANOVA and did not revealed any significant differences in T2 signal between days for NMR-assay with casein-coated nanoparticles. We suppose that these slight differences are by chance. For gelatin-coated nanoparticles we observed significant growth of T2 in zero samples between days (Fig S13), however we are unable to explain them. All measurements were done by the same operator, all reagents were from the same batches.

Reviewer 2 Report

The submitted manuscript entitled Magnetic nanoclusters coated with albumin, casein, and gelatin: size tuning, relaxivity, stability, protein corona and application in nuclear magnetic resonance immunosensing by Khramtsov et al. reports the development and physico-chemical characterization of nanoclusters of magnetic iron-carbon nanoparticles covered with four-types of proteins namely bovine serum albumin (BSA), casein, gelatins A and B, respectively. Authors evaluated the effect of pH, ionic strength, protein-to-nanoparticle mass ratio and sonication time on the protein-coated nanoclusters formation. The aminated iron-carbon nanoparticles (Fe@C-NH2) were conjugated with Streptavidin (Fe@C-NH2/Protein/Str) and Protein G (Fe@C-NH2/Protein/G). The Fe@C-NH2/Protein/Str nanoclusters were characterized by means of microscopic (AFM, TEM), spectroscopic (UV-VIS) and TGA analysis. The Fe@C-NH2/Protein/G nanoclusters (conjugated with protein G) were employed in the NMR-immunosensing of IgG against the anti-tetanus toxoid. While the material part seems to be well structured and written with plenty experimental data, the part relating to materials and discussion is charged with too many graphs, sometimes, underlining the same information and further making the manuscript difficult to be followed. More than that, the part related to NMR-based immunosensors should be better emphasized/ documented. Several issues should be sorted out by the authors before the manuscript publication.

The paper needs MINOR corrections. My comments are

Major aspects:

A table containing the analytical parameters (linear range, limit of detection (LOD), sensitivities etc) of the proposed NMR-assay for determining anti-tetanus antibodies as well as a comparison with the data reported previous in literature should be provided. A consisted selection of the graphs should be performed all over the manuscript. The manuscript is overloaded with too many graphs telling the same information. For instance: i) The graphs in Fig. 2 of the manuscript are same as the graphs provided in Figure S3, Figure S4 and Figure S5, in supplementary info. ii) In Figure 3 is enough to present only Fig.3(d) which is a summary of Fig.3(a), Fig.3(b) and Fig.3(c). etc

Minor aspects:

“the protein coronas of nanoclusters were studied” Please provide the type of the study. define the terms T2( (line 49, 52 etc) and r2 (line 27, 76 etc). in vivo (line 48, 60 etc); in vitro (line 49, 50, 60, 74 etc) should be write italic all over the manuscript part 3.6 is missing from Results and discussion part

Author Response

Dear Editor-in-Chief and Referees

Authors are very grateful for your comments to our manuscript and thoughtful suggestions. Based on these comments and suggestions, we have made modifications to the original paper. We believe that the manuscript has been greatly improved and hope it has reached your journal's standard. Please, find our answers to yours comments below. Added or changed pieces of text are highlighted with yellow color.

We changed the title of the article because the NMR-assay is not sensor, but rather a dot-immunoassay with disposable test-strips and NMR detection.

Comment

Major aspects:

A table containing the analytical parameters (linear range, limit of detection (LOD), sensitivities etc) of the proposed NMR-assay for determining anti-tetanus antibodies as well as a comparison with the data reported previous in literature should be provided.

Response

A table on comparison of the NMR-assay with previously described techniques of anti-tetanus IgG detection was added to the section 3.4. (Table 2)

References 43-48 were added. Lines 554-557 were added.

Comment

A consisted selection of the graphs should be performed all over the manuscript. The manuscript is overloaded with too many graphs telling the same information. For instance: i) The graphs in Fig. 2 of the manuscript are same as the graphs provided in Figure S3, Figure S4 and Figure S5, in supplementary info. ii) In Figure 3 is enough to present only Fig.3(d) which is a summary of Fig.3(a), Fig.3(b) and Fig.3(c). etc

Response

We made corrections according to the comments of the reviewer.  Figure 2 was removed as well as parts a, b and c of Fig. 3. Figure 3 was renamed to “Figure 2”.

Comment

Minor aspects:

“the protein coronas of nanoclusters were studied” Please provide the type of the study. define the terms T2( (line 49, 52 etc) and r2 (line 27, 76 etc). in vivo (line 48, 60 etc); in vitro (line 49, 50, 60, 74 etc) should be write italic all over the manuscript part 3.6 is missing from Results and discussion part

Response

We made corrections according to the comments of the reviewer.  Terms r2 and T2 were defined (line 21 and 53-54). In vivo and in vitro have been written in italics. Numeration of sections was corrected.

Round 2

Reviewer 1 Report

This revised manuscript seems to be much better for me.

Reviewer 2 Report

The manuscript is suitable for publication in the present form.